# Optimization of Sustainable Bi-Objective Cold-Chain Logistics Route Considering Carbon Emissions and Customers' Immediate Demands in China

**Zhichao Ma, Jie Zhang, Huanhuan Wang * and Shaochan Gao**

School of Management Engineering and Business, Hebei University of Engineering, Handan 056038, China
* Correspondence: wanghuanhuan202302@126.com

**Abstract:** To meet the national green development trend and realize the sustainable development of human society, the carbon emission in cold-chain distribution is costed. We plan the vehicle distribution path reasonably and optimize the distribution path locally for immediate demand to balance the economic benefits of enterprises and customer satisfaction while reducing the environmental pollution. To minimize distribution cost and maximize customer satisfaction, we design an improved ant colony algorithm to solve the initial distribution path and use the insertion method to solve the immediate customer demand. Taking the actual data of enterprise M as an example, we obtain the optimal distribution path using MATLAB software and optimize the distribution path locally according to the immediate customer demand. The results show that the proposed model and the designed algorithm are practical in satisfying the sustainable development of cold-chain logistics in China.

**Keywords:** cold-chain logistics; carbon emissions; immediate needs of customers; improved ant colony algorithm





## 1. Introduction

With the development of the economy and the improvement of people's living standards, the quantity and quality of fresh products are increasing, and the development of cold-chain logistics is also very rapid. However, rapid development has also brought high costs and high emissions. In the context of the Chinese "3060" policy of peaking carbon neutrality, finding the balanced development of the economy, energy, and environment is an essential issue for China and its provinces, cities, and other economies. Cold-chain logistics distribution often involves fresh products. Perishability refers to abnormal quality problems such as the death, deterioration of animal food, or the decay and mildew of plant food. In cold-chain transportation and storage, businesses will reduce the temperature and other means to prevent the deterioration of fresh products. However, such measures usually result in carbon emissions of cold-chain logistics exceeding the standard. Therefore, it is essential to consider carbon emissions in cold-chain logistics distribution. This scenario creates an immediate demand from the customer. With the development of the customer's individualized and differentiated needs, the original Business to Customer (B2C), Customer to Customer (C2C) cold-chain logistics model cannot meet the immediate needs of customers. Therefore, in cold-chain logistics, distribution path optimization considers customers' immediate needs in line with the current consumption trends. Whether the customer's immediate demand is satisfied or not can be reflected in the customer's satisfaction, and the logistics cost directly affects the economic benefit of the enterprise. Currently, research on cold-chain logistics focuses on customer satisfaction and route optimization, and few articles consider both carbon emissions and immediate customer needs. In this paper, a bi-objective optimization model aiming at customer satisfaction cost and logistics cost is established.

The research shows that the Asia–Pacific region is the primary cold-chain logistics market, and China contributes the most to generating market income. The Chinese cold-chain logistics market has grown 15 percent annually since 2013 and is expected to generate USD 80 billion in 2024 revenues [1]. Although the cold-chain logistics market is booming, cold-chain logistics companies are usually small and numerous. Cooperation between them is limited, resulting in high transport costs and carbon emissions [2,3]. Therefore, reducing total distribution costs and carbon emissions is a significant concern for the industry involving cold-chain logistics companies. Carbon emissions become an essential factor affecting the distribution path in the vehicle-routing problem. In order to better consider the carbon emission factor in the model, some scholars add the carbon emission cost into the objective function [4,5].

In the research of path optimization, there are three ways to reduce carbon emissions: carbon tax regulation, carbon trading regulation, and multi-objective optimization. Carbon tax regulations exist in many places, such as in the United States, where companies do not receive subsidies or tax credits to remanufacture and, therefore, must pay an emissions tax [6]. Therefore, green cold-chain logistics has become the future trend of development. Based on this, Dou and others believe that collecting second-hand products directly from customers can effectively reduce the carbon emissions of the supply chain system. So, they modeled three classes. The results show that manufacturers and retailers can be the most eco-efficient recycling channels, while third-party recycling channels are the least popular. Retailers can do a better job of Pareto efficiency both environmentally and economically [7].

The tax rate in the carbon tax regulations may change from time to time. The results show that the total emission can increase or decrease by reducing remanufacturing emission intensity. In order to effectively control total emissions, regulators can selectively increase tax rates according to the characteristics of manufacturers' production decisions and remanufacturing [8]. In order to promote low-carbon production, the government also provides some subsidies under the carbon tax regulations. There are two main types of subsidies: those based on emission reductions and the cost of investment in reducing emissions. The results show that government subsidies can expand the corresponding conditions for improving investment in emission reduction [9]. To illustrate the impact of various carbon footprint plans on costs and carbon emissions, we have developed containerization strategies to minimize transport costs under different carbon footprint plans. The study shows that the containerization strategy under the carbon tax regulation is better than the conventional policy regarding the total transport cost and carbon emissions [10].

The researchers regulated carbon trading by calculating the cost of carbon and introduced carbon trading by focusing on carbon prices and carbon quotas. If a cold-chain logistics company emits more carbon than it can, it must pay extra to buy more. However, if the emissions of the cold-chain logistics company are below the prescribed limit, the carbon quota can be sold for a profit [11]. Research shows that as the price of carbon rises, so does the cost of carbon. As carbon becomes more expensive than before, emissions significantly affect the total cost [12]. When carbon quotas are fixed, the higher the price of carbon, the greater the total cost of carbon [13]. In order to study the relationship between the fluctuation of the carbon price and production efficiency and trading quantity, modeling simulation and sensitivity analysis were carried out. The results show that under carbon cap-and-trade regulation, joint production and trading policies can help firms benefit from changes in carbon prices [14]. In order to study the relationship between the carbon price and enterprise marginal emission cost, the time-varying difference (TDID) model is used to post-estimate the carbon price of enterprises. The results show that the impact of carbon prices is more pronounced among state-owned enterprises in eastern China. In addition, more significant investment in R&D patents and capital can help raise corporate total factor productivity [15]. The impact of carbon trading regulation on different enterprises' green innovation is heterogeneous. The behavior logic of participants under the regulation of carbon trading is analyzed by constructing an evolutionary game, evolutionary equilibrium, and numerical analysis. The results show that when the impact of carbon trading

regulation on innovation is positive, the quality of innovation may be different under different conditions [16,17].

In addition, there are related researchers through multi-objective optimization methods to reduce carbon emissions. For example, a multi-objective mixed-integer linear programming model calculates the amount of greenhouse gas emissions related to the minimum cumulative energy consumption, thus providing a decision-making basis for reducing carbon emissions. The study shows that the proposed model can reduce greenhouse gas emissions from offshore by 25%. Therefore, the shared power generation between offshore wind farms and platforms is beneficial to the environmental and economic benefits of society and enterprises [18]. The combination of multiple modes of transport provides a flexible and environmentally friendly alternative for transporting large quantities of goods over long distances. In order to reflect the advantages of each mode of transport, the multi-objective planning of multimodal transport research shows that the cost of excessive reduction in carbon emissions and the original cost of transport are similar [19]. In order to study the effect of supply chain structure on carbon emission, a new network optimization model is proposed. Research shows that the rate of change in carbon emissions is related to the cost of carbon emissions and GDP growth. Countries with higher carbon costs will bear more burden [20]. In order to avoid increasing energy consumption in developed countries due to enhanced manufacturing technology and mitigate global warming, the supply chain network optimization was carried out to minimize the total production cost, reduce the carbon footprint, and minimize the energy cost of renewable energy. The results show that the optimized supply chain has lower production costs and less carbon emission [21].

This article aims to calculate the cost of carbon emissions to conduct carbon tax regulation and then control carbon emissions. The distribution cost of the vehicle-routing model in this paper includes vehicle fixed cost, vehicle transportation cost, temperature cost, and carbon emission cost. The cost of carbon emissions is the cost of carbon dioxide emissions mainly from the following two aspects. On the one hand, refrigerated vehicles produce carbon emissions from fuel consumption; on the other hand, refrigerant consumption by refrigeration equipment to keep the products of the cold chain in a suitable low-temperature environment during transportation also produces carbon emissions. Therefore, the carbon emission cost is considered in the vehicle-routing model.

In the current cold-chain logistics research, many articles consider carbon emission and logistics cost, but they consider carbon emission, logistics cost, and immediate customer demand less. This paper takes the carbon emission cost as a part of the logistics cost. It takes the logistics cost and the customer's immediate demand as two goals to optimize the cold-chain logistics distribution path. Based on these two factors, a bi-objective optimization model for customer satisfaction and logistics costs is established in this paper. The trapezoidal fuzzy membership function was constructed to express the relationship between customer satisfaction and the time window and to optimize the local allocation path. This paper also considers optimizing the local allocation of the total cost of carbon emissions. The research could help increase customer satisfaction and reduce carbon emissions from cold-chain logistics. It is of great practical significance to realize the maximization of environmental and economic benefits. In addition, this study adapts to the development of the current era and meets the requirements of low-carbon sustainable development of cities. The heuristic function and pheromone in the model optimization process are improved, and multi-strategy improvement is added to the ant colony algorithm. The effectiveness of the algorithm and the proposed model considering the cost and customer satisfaction of emission reduction is verified by an example.

The innovation of this article is as follows. First, this paper focuses on customer demand and carbon emissions and establishes a bi-objective optimization model with customer satisfaction and logistics costs as objectives. Second, the cost of carbon emission is considered calculating logistics costs. This factor is rare in previous studies. The model helps to strike a balance between customer satisfaction, business economic benefits, and environmental benefits. Third, the ant colony algorithm is improved. In this paper, the

heuristic function and pheromone are improved, and the multi-strategy improvement is added to the ant colony algorithm to form an improved ant colony algorithm to solve the initial distribution path.

This paper is organized as follows: Section 2 contains the problem description and modeling. The total cost of distribution, including the cost of carbon emissions, was constructed to achieve both the lowest cost of distribution and the best customer satisfaction, taking into account customers' needs. Section 3 explains the algorithm design. We use the improved ant colony algorithm to solve the distribution path in the initial stage. Moreover, the insert method solves the customer's instant demand. An example analysis of the calculation is introduced in Section 4. This paper uses a sizeable fresh distribution company in Shaanxi province as an example to calculate. Finally, Section 5 contains the conclusion.

## 2. Problem Description and Modeling

### 2.1. Problem Description

The cold-chain logistics enterprise has a group of customers and a distribution center, and the refrigerated trucks serve the customers under the premise of meeting the vehicle load limit. The distribution center has sufficient inventory, and the distribution center has the same vehicle model and a sufficient number of distribution vehicles. The distribution center needs to make path planning for the known customer demand, which is the initial distribution route. In the process of distribution, immediate customer demand will be generated. Then, the cold-chain logistics enterprise needs to handle the immediate customer demand and adjust the original distribution route formulated. Because of the perishable nature of cold-chain products, the refrigeration equipment in the distribution process will consume more energy and increase emissions. Therefore, in this paper, the total distribution cost, including the carbon emission cost, is constructed while considering the immediate customer demand to achieve the two goals of the lowest distribution cost and the most excellent customer satisfaction. The schematic diagram of vehicle path optimization under immediate customer demand is shown in Figure 1.

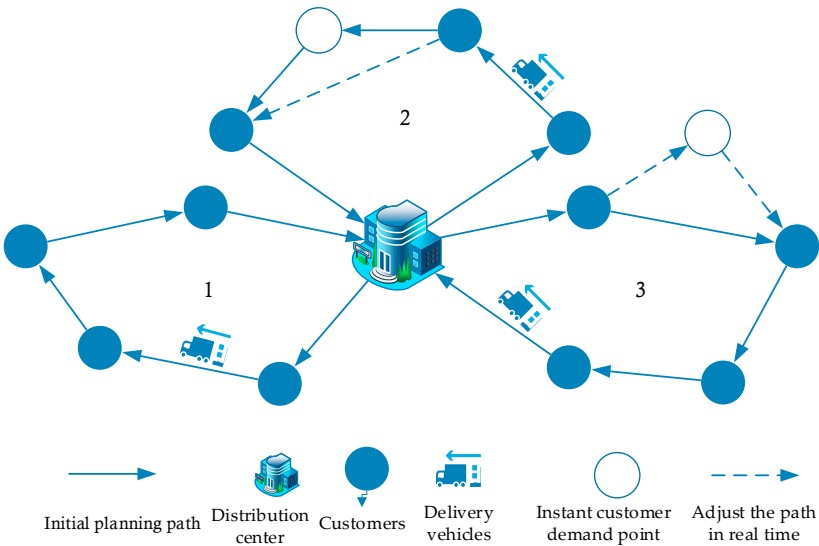

**Figure 1.** Dynamic vehicle route optimization diagram.

The assumptions of this paper are as follows.

**Supposition 1:** *There is only one distribution center, and distribution vehicles must start and finish the routes.*

**Supposition 2:** *The number of distribution vehicles is sufficient and of the same type, and the total amount of goods loaded in the distribution vehicles cannot exceed their capacity.*

**Supposition 3:** *Customer demand cannot be split, and each customer point is served.*

**Supposition 4:** *Distribution vehicles are loaded before distribution, and they do not need to wait when they arrive at any customer demand point and can be unloaded directly.*

**Supposition 5:** *Distribution vehicles operate at a constant speed.*

**Supposition 6:** *All drivers of distribution vehicles have the same technical experience, and vehicle fuel consumption will not change due to subjective factors.*

*2.2. Distribution Cost Function*

(1) Vehicle fixed cost

For logistics enterprises, as long as the use of distribution vehicles to pay the fixed cost of vehicle use, fixed costs and a load of goods and the length of the driving distance is not relevant:

$$C_1 = \sum_{j=1}^{n} \sum_{k=1}^{m} f_k x_{0j}^k \tag{1}$$

(2) Vehicle Transportation Costs

Transportation costs mainly include fuel consumption, vehicle maintenance, and other costs. Generally speaking, vehicle transportation costs increase with the distance traveled:

$$C_2 = \sum_{i=0}^{n} \sum_{j=0}^{n} \sum_{k=1}^{m} F_k x_{ij}^k d_{ij} \tag{2}$$

(3) Temperature cost

The temperature cost includes the temperature generated when the refrigerated truck is driving. Moreover, the temperature cost is caused by the additional consumption of refrigerant. Moreover, the reason is the temperature difference between the refrigerated truck compartment and the external environment in the loading and unloading process, generating air convection and raising the temperature inside the refrigerated truck compartment.

The temperature cost in the transportation process is calculated as follows:

$$C_{31} = \sum_{i=0}^{n} \sum_{j=0}^{n} \sum_{k=1}^{m} x_{ij}^k \phi_1 t_{ij}^k T \tag{3}$$

The formula calculates the temperature cost of the loading and unloading link:

$$C_{32} = \sum_{i=0}^{n} \sum_{j=0}^{n} \sum_{k=1}^{m} x_{ij}^k \phi_2 t_j^k \Delta T \tag{4}$$

In summary, the temperature cost expression is:

$$C_3 = C_{31} + C_{32} = \sum_{i=0}^{n} \sum_{j=0}^{n} \sum_{k=1}^{m} x_{ij}^k (\phi_1 t_{ij}^k T + \phi_2 t_j^k \Delta T) \tag{5}$$

All the symbols in the text are shown in Table 1.

**Table 1.** Symbolic conventions for the proposed model.

| Variable Classification | Symbol | Connotation and Unit |
|---|---|---|
| Parametric variable | $K$ | A collection of vehicles providing fresh produce delivery services, $k \in K$ and $k = (1, 2, \ldots m)$; |
| | $I$ | The collection of distribution points where transportation is required. $i, j \in I$ and $i, j = (0, 1, 2 \ldots n)$, where 0 is the distribution center, and the others are the demand points; |
| | $f_k$ | Fixed cost of using any of the delivery vehicles; |
| | $F_k$ | Transportation cost per unit distance; |
| | $d_{ij}$ | The distance between client $i$ and client $j$; |
| | $t_{ij}^k$ | The time it takes for the $k$ vehicle to travel from distribution point $i$ to distribution point $j$; |
| | $\phi_1$ | The cost factor for product cooling during distribution; |
| | $T$ | High and low temperatures in the carriage; |
| | $t_j^k$ | Length of time required for the vehicle $k$ to load and unload at the distribution point $j$; |
| | $\phi_2$ | The cost factor for product cooling during loading and unloading; |
| | $\Delta T$ | The existence of temperature differences between the inside and outside of the carriage; |
| | $Q$ | The load limit of the vehicle; |
| | $q_i$ | Customer demand $i$; |
| | $Q_{ij}$ | Vehicle shipment from customer $i$ to customer $j$; |
| | $p_c$ | Unit carbon tax price; |
| | $e$ | Carbon dioxide emission factor; |
| | $\lambda_i$ | The actual time the delivery vehicle arrives at the customer demand point; |
| | $C_e$ | Waiting cost per unit time advance service; |
| | $C_l$ | Penalty cost per unit of time delayed service; |
| Collection variable | $[et_i, lt_i]$ | Optimal service time for the client $i$; |
| | $[ET_i, LT_i]$ | A soft time window limit for the client $i$; |
| Decision variable | $x_{ij}^k \begin{cases} 1, \text{Distribution } k \text{ vehicles travels from customer } i \text{ to customer } j \\ 0, \text{Other} \end{cases}$ | |

(4) Cost of carbon emissions

The carbon emission cost studied in this paper is the cost of carbon dioxide emission, which is generated mainly from the following two aspects. On the one hand, fuel consumption fuel by refrigerated trucks generates carbon emissions; on the other hand, refrigerant by refrigeration equipment also generates carbon emissions to keep the cold-chain products in a suitable low-temperature environment during transportation.

When a cargo of weight $Q_{ij}$ is loaded in the reefer, the fuel consumption per unit distance is calculated using the load estimation method as shown in Equation (6):

$$E_1(Q_{ij}) = \rho^0 + \frac{\rho^* - \rho^0}{Q} Q_{ij} \tag{6}$$

where $\rho^0$ is the fuel consumption rate per unit distance when the vehicle is empty, and $\rho^*$ is the fuel consumption rate per unit distance when the vehicle is fully loaded.

The cost of carbon emissions from vehicle operation is shown in Equation (7) [16]:

$$C_{41} = p_c \sum_{i=0}^{n} \sum_{j=0}^{n} \sum_{k=1}^{m} x_{ij}^k e E_1(Q_{ij}) d_{ij} \tag{7}$$

The refrigeration method of the refrigerated vehicle in this paper is mechanical refrigeration. Therefore, it could consume diesel or gasoline to maintain the cold-chain products at a lower temperature while generating carbon dioxide. The cost of carbon emissions from

the refrigeration equipment for a vehicle carrying a product of weight to be delivered from customer to customer is shown in Equation (8):

$$C_{42} = p_c \sum_{i=0}^{n} \sum_{j=0}^{n} \sum_{k=1}^{m} x_{ij}^k E_2 Q_{ij} d_{ij} \tag{8}$$

In summary, the expression for the cost of carbon emissions is:

$$C_4 = C_{41} + C_{42} = p_c \sum_{i=0}^{n} \sum_{j=0}^{n} \sum_{k=1}^{m} x_{ij}^k d_{ij} (eE_1(Q_{ij}) + E_2 Q_{ij}) \tag{9}$$

(5) Time window penalty cost

The time window penalty cost is the extra cost of the cold-chain delivery vehicle for violating the time window specified by the customer. The time window penalty cost includes the opportunity cost of forming a window earlier than the time window and the penalty cost of forming a window later than the time window. A fuzzy soft time window is used in this paper. Because the best service time is not the whole time window but a specific period within the time window, the traditional soft time window cannot accurately express customer satisfaction, so the fuzzy soft time window is chosen. Assuming a customer $i$ service start time of $\lambda_i$, the time window penalty function can be expressed as follows:

$$C_5(i) = \begin{cases} C_e(ET_i - \lambda_i), & \lambda_i < ET_i \\ 0, & ET_i \leq \lambda_i \leq LT_i \\ C_l(\lambda_i - LT_i), & \lambda_i > LT_i \end{cases} \tag{10}$$

The customer's $i$ time window penalty is shown schematically in Figure 2.

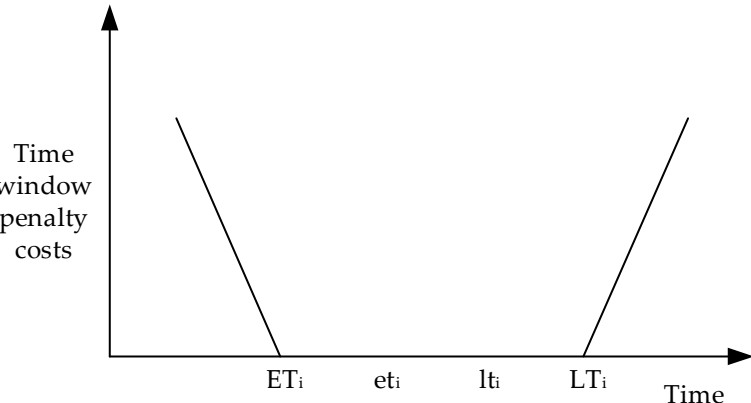

**Figure 2.** Time window punishment diagram.

Therefore, the customer $i$ time window costing is expressed as follows:

$$C_5 = C_e \sum_{i \in I} \max\{ET_i - \lambda_i, 0\} + C_l \sum_{i \in I} \max\{\lambda_i - LT_i, 0\} \tag{11}$$

where $C_e \sum_{i \in I} \max\{ET_i - \lambda_i, 0\}$ denotes the total opportunity cost due to being earlier than the time window and $C_l \sum_{i \in I} \max\{\lambda_i - LT_i, 0\}$ is the total penalty cost due to being later than the time window.

From the above analysis, the total distribution cost objective function can be obtained as follows:

$$C = C_1 + C_2 + C_3 + C_4 + C_5 \tag{12}$$

i.e.,

$$C = \sum_{j=1}^{n}\sum_{k=1}^{m} f_k x_{0j}^k + \sum_{i=0}^{n}\sum_{j=0}^{n}\sum_{k=1}^{m} F_k x_{ij}^k d_{ij} + \sum_{i=0}^{n}\sum_{j=0}^{n}\sum_{k=1}^{m} x_{ij}^k(\phi_1 t_{ij}^k T + \phi_2 t_j^k \Delta T)$$
$$+ p_c \sum_{i=0}^{n}\sum_{j=0}^{n}\sum_{k=1}^{m} x_{ij}^k \cdot d_{ij}(eE_1(Q_{ij}) + E_2 Q_{ij}) + C_e \sum_{i \in I} \max\{ET_i - \lambda_i, 0\} \qquad (13)$$
$$+ C_l \sum_{i \in I} \max\{\lambda_i - LT_i, 0\}$$

### 2.3. Customer Satisfaction Function

In the actual distribution process of cold-chain logistics, the customer time window is flexible, and customer satisfaction also decreases with the increase in the difference between the actual arrival time of the distribution vehicle and the preset time of the customer. Customer satisfaction based on a soft time window is generally transformed into 0–1 values, where 0 and 1 denote the minimum and maximum satisfaction, respectively. The relationship between soft time windows and customer satisfaction is illustrated in Figure 3.

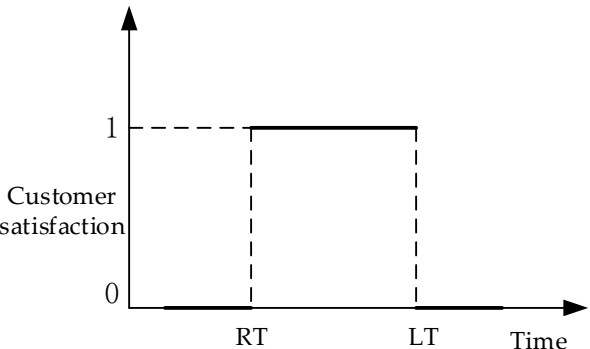

**Figure 3.** Relationship between soft time window and customer satisfaction.

However, because the customer's best service in this paper is only part of the customer time window—a specific period—the traditional soft time window cannot accurately represent customer satisfaction. This paper's trapezoidal fuzzy affiliation function is constructed to represent the time window, as shown in Figure 4.

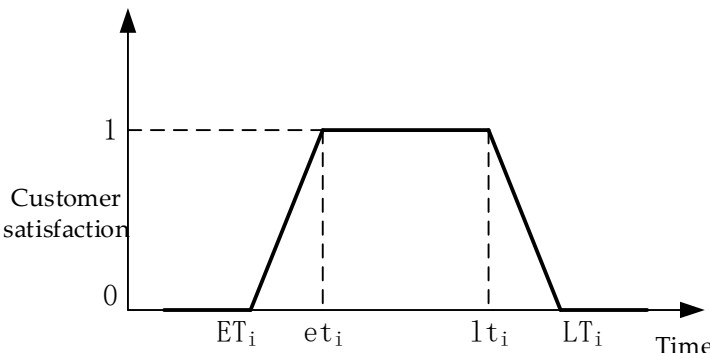

**Figure 4.** Relationship between customer satisfaction and time window of trapezoidal membership function.

The function of customer satisfaction $s_i$ is expressed as follows:

$$s_i = \begin{cases} \frac{\lambda_i - ET_i}{et_i - ET_i}, & ET_i \leq \lambda_i < et_i \\ 1, & et_i \leq \lambda_i < lt_i \\ \frac{LT_i - \lambda_i}{LT_i - lt_i}, & lt_i \leq \lambda_i < LT_i \\ 0, & Other \end{cases} \tag{14}$$

So, the customer satisfaction function can be expressed as:

$$S = \frac{\sum\limits_{i \in I} s_i}{n} \tag{15}$$

### 2.4. Mathematical Models

This model is nonlinear, and the improved ant colony algorithm has vital positive feedback and parallelism to solve this model. In summary, in order to unify the calculation and transform the maximum customer satisfaction into the minimum customer dissatisfaction, the optimization model of the cold-chain logistics distribution path considering carbon emission and immediate customer demand is established as follows:

$$\begin{aligned} minC= &\sum_{j=1}^{n}\sum_{k=1}^{m} f_k x_{0j}^k + \sum_{i=0}^{n}\sum_{j=0}^{n}\sum_{k=1}^{m} F_k x_{ij}^k d_{ij} + \sum_{i=0}^{n}\sum_{j=0}^{n}\sum_{k=1}^{m} x_{ij}^k (\phi_1 t_{ij}^k T + \phi_2 t_j^k \Delta T) \\ &+ p_c \sum_{i=0}^{n}\sum_{j=0}^{n}\sum_{k=1}^{m} x_{ij}^k d_{ij}(eE_1(Q_{ij}) + E_2 Q_{ij}) + C_e \sum_{i \in I} \max\{ET_i - \lambda_i, 0\} \\ &+ C_l \sum_{i \in I} \max\{\lambda_i - LT_i, 0\} \end{aligned} \tag{16}$$

$$minS = 1 - \frac{\sum\limits_{i \in I} s_i}{n} \tag{17}$$

The constraints are as follows:

$$\sum_{i=0}^{n} q_i \sum_{j=0}^{n} x_{ij}^k \leq Q, k \in \{1, 2, \ldots m\} \tag{18}$$

$$\sum_{j=0}^{n}\sum_{k=1}^{m} x_{ij}^k = 1, i \in \{1, 2, \ldots, n\} \tag{19}$$

$$\sum_{i=0}^{n}\sum_{k=1}^{m} x_{ij}^k = 1, j \in \{1, 2, \ldots, n\} \tag{20}$$

$$\sum_{j=1}^{n} x_{ij}^k = \sum_{j=1}^{n} x_{ji}^k \leq 1 \tag{21}$$

$$\sum_{i,j \in S \times S}^{j} x_{ij}^k \leq |S| - 1, S \subseteq \{1, 2 \ldots n\} \tag{22}$$

$$x_{ij}^k \begin{cases} 1, Distribution\ k\ vehicles\ travels\ from\ customer\ i\ to\ customer\ j \\ 0, Other \end{cases} \tag{23}$$

Equation (16) indicates the lowest total cost of distribution. Equation (17) indicates the lowest customer dissatisfaction. Equation (18) indicates that the load capacity of each distribution vehicle does not exceed its capacity. Equations (19) and (20) indicate that each distribution point will be served, and only one distribution vehicle can provide a distribution service. Equation (21) indicates that each distribution vehicle must start and

end at the distribution center. The purpose of Equation (22) is to eliminate the subloop. Equation (23) represents the 0–1 variable of the distribution vehicle *k* from point *i* to point *j*.

The objective functions in this paper are the lowest total cost of distribution and the lowest customer dissatisfaction, which need to be quantified. In this paper, the sigmoid function is used to unify the order of magnitude of each objective function so that the order of magnitude of the two objective function values remains unified, and the quantified results are:

$$(\min C)' = 1/(1 + e^{-\left(\sum\limits_{j=1}^{n}\sum\limits_{k=1}^{m} f_k x_{0j}^k + \sum\limits_{i=0}^{n}\sum\limits_{j=0}^{n}\sum\limits_{k=1}^{m} F_k x_{ij}^k d_{ij} + \sum\limits_{i=0}^{n}\sum\limits_{j=0}^{n}\sum\limits_{k=1}^{m} x_{ij}^k (\phi_1 t_{ij}^k T + \phi_2 t_j^k \Delta T) + p_c \sum\limits_{i=0}^{n}\sum\limits_{j=0}^{n}\sum\limits_{k=1}^{m} x_{ij}^k d_{ij}(eE_1(Q_{ij}) + E_2 Q_{ij}) + C_e \sum\limits_{i\in I} \max\{ET_i - \lambda_i, 0\} + C_l \sum\limits_{i\in I} \max\{\lambda_i - LT_i, 0\}\right)}) \quad (24)$$

$$(\min S)' = 1/(1 + e^{-(1 - \frac{\sum\limits_{n\in I} S_n}{n})}) \quad (25)$$

The objective function value $y$ is obtained after the normalization process $y'$, $y' \in [0,1]$. $C'$ and $S'$ represent $C$ and $S$ after the above method.

## 3. Algorithm Design

The objective function of this paper is the minimum cost and the maximum customer satisfaction, which belongs to the multi-objective optimization problem and is usually solved by the heuristic. Although in the initial stage, the ant colony algorithm is prone to fall into local optimum and slow convergence because of the lack of pheromones, it can be improved to make up for the algorithm's shortcomings. Therefore, this paper improves the traditional ant colony algorithm, using the improved ant colony algorithm to solve the initial stage of the distribution path. In the stage of customer demand, the insert method is simple and efficient, and the quality of a solution is generally high, so the insert method is chosen to solve customer demand.

Macro Dorigo first proposed the ant colony algorithm, which has vital positive feedback and parallelism [22]. Although the ant colony algorithm is prone to fall into local optimum and slow convergence in the initial stage because of the lack of pheromones, it can be improved to make up for the algorithm's shortcomings [23]. Therefore, in this paper, the traditional ant colony algorithm is improved, and the distribution path in the initial stage is solved using the improved ant colony algorithm. In the immediate customer demand stage, the insert method is simple and efficient, and the solution quality is generally high, so the insertion method is chosen to solve the immediate customer demand [24].

### 3.1. Algorithm for Initial Distribution Path Planning

3.1.1. Principle of Ant Colony Algorithm

The ant colony algorithm has positive feedback and robustness to obtain a better solution. From nest to food, the trajectory of an ant is disorderly, but research has found that when the number of ants accumulates to a specific value, they always find the shortest path. When ants search for food, they release a secretion—pheromone—on their path. When an ant encounters an intersection, it randomly chooses a path. When the next ant comes to the same intersection, it chooses the route with a higher concentration of pheromones and releases pheromones. The pheromone concentration increased as the distance became shorter, and this affected the path choice of the following ants. As time passes and the number of ants accumulates, eventually, all ants will choose the path with the greater pheromone concentration, i.e., the shortest path, which also finds the optimal solution to the problem.

3.1.2. Improved Ant Colony Algorithm

(1)    Adopt a saving matrix to guide the ant search

The basic ant colony algorithm is prone to fall into local optimum, and to solve this phenomenon, this paper adds the saving matrix $U : U(i,j) = D(i,1) + D(j,1) - D(i,j)$ to strengthen the attractiveness to ants [22]. After optimization using the idea of the saving matrix, the original distribution route is merged from two to one, at which time the probability is shown in Equation (26):

$$P_{ij}^k(t) = \begin{cases} \dfrac{[\tau_{ij}(t)]^\alpha \times [\eta_{ij}(t)]^\beta \times U_{ij}^\theta}{\sum\limits_{s \in allowed_k} [\tau_{is}(t)]^\alpha \times [\eta_{is}(t)]^\beta \times U_{is}^\theta}, j \in allowed_k \\ 0, \qquad\qquad\qquad j \notin allowed_k \end{cases} \tag{26}$$

$\theta$ is the weighting factor that can be adjusted for the savings matrix.

(2)    Improvement of the heuristic function

This is a heuristic article. The heuristic is presented relative to the optimal algorithm, and a feasible solution of the combinatorial optimization problem to be solved is given within an acceptable range. The deviation between the feasible solution and the optimal solution cannot be predicted; when dealing with large-scale VRP, the heuristic is more feasible. Pheromones are a kind of volatile secretion released by ants to the environment, which will disappear gradually with the passage of time. Although in the initial stage, the ant colony algorithm is prone to fall into local optimum and slow convergence because of the lack of pheromones, it can be improved to make up for the shortcomings of the algorithm. In this paper, the heuristic function and pheromone are improved by using an econometric matrix to guide the ant search. The multi-strategy improvement of sequential exchange strategy [25], 2-OPT algorithm [26] and sequential insertion strategy [27] are added into the ant colony algorithm to strengthen the aim of ant searching, avoid the local optimum and finally reach the global optimum [28]. The expression is shown in Equation (27):

$$\eta_{ij}(t) = \frac{1}{d_{ij} + d_{ig}} \tag{27}$$

where $d_{ig}$ indicates the distance between the current nodes $j$ and $g$.

(3)    Improvement of pheromone

Since the chances of each path being chosen are the same in the initial moment environment, this leads to the time for ants to find the shortest path length. In order to solve the problem in which ants are prone to fall into local optimum, this paper improves the initial pheromone by using chaotic variables, which are made to correspond to the optimization variables when the pheromone is initialized. The chaotic variables are generated by the logistic mapping of

$$Z_{ij}(t+1) = \mu Z_{ij}(t)\big[1 - Z_{ij}(t)\big] \tag{28}$$

In the above equation, $\mu$ is the control variable, which takes the value between [3.56, 4.0] in general, and $Z_{ij}(t)$ is generated randomly [29]. When, $\mu = 4, 0 \le Z_{ij}(0) \le 1$ is fully generated randomly. The purpose of the chaotic system is to make the same initial pheromone chaotic in the essential ant colony to reduce the occurrence of local optima, make it more diverse, and finally improve the convergence speed of the algorithm.

Although chaotic systems can chaos the initial pheromone, they cannot avoid the occurrence of the ant colony algorithm falling into a local optimum. Therefore, the pheromone

is adjusted and optimized to reduce instances of the algorithm falling into a local optimum. The expression after incorporating chaotic perturbations in the pheromone is:

$$
\begin{aligned}
\tau_{ij}(t+1) &= (1-\rho)\tau_{ij}(t) + \Delta\tau_{ij}(t) + \zeta Z_{ij}(t) \\
\Delta\tau_{ij}(t) &= \sum_{k=1}^{n} \tau_{ij}^{k}(t) \\
\Delta\tau_{ij}^{k}(t) &= \begin{cases} Q/L_k & \text{The } k \text{ ant moves from location } i \text{ to location } j \\ 0 & else \end{cases}
\end{aligned}
\tag{29}
$$

where $Z_{ij}(t)$ is the chaotic variable obtained from the iteration of Equation (29).

(4)  Multi-strategy improvement This paper uses a multi-strategy approach to adjust the solutions obtained from each iteration. Moreover, we add a sequential exchange strategy [25], 2-OPT algorithm [26], and a sequential insertion strategy to the ant colony algorithm [27], in which each strategy is a neighborhood to avoid local optima to enhance the ergodicity of the ant colony algorithm search.

  (1)  Sequential exchange strategy: Each customer point is passed through in sequence, and a customer point within the current line is exchanged with a customer on the same line or another line for the location.
  (2)  The 2-OPT algorithm: Two points on the route are randomly selected, and the order of the remaining points remains the same, only the points between them are flipped in reverse order, which belongs to a local search algorithm.
  (3)  Sequential insertion strategy: Insert customer points in different routes.

### 3.1.3. Improved Ant Colony Algorithm Flow

(1)  Initialization parameters. Let time $t = 0$, iteration number $iter = 0$, set maximum iteration number $iter_{\max}$, input specific data such as the distribution center and customer geographic location, set the distribution center node as the starting point of the ant, and enable chaos initialization.
(2)  Under the restriction of satisfying multiple constraints, each ant selects the next node $j$ according to Equation (26), records it in the forbidden table, and updates the load information of the vehicle.
(3)  Determine whether all ants visit all customer points. If not, repeat the step; if yes, put all customer points into the forbidden table and return to the logistics distribution center.
(4)  Multi-strategy improvement is performed.
(5)  Update the pheromone using the pheromone update rule Formula (29) of chaotic perturbation.
(6)  When the number of loops reaches the set maximum number of loops, the algorithm is terminated, and the optimal result of the algorithm is output. Conversely, the taboo table is emptied, and a new round of loops is started, $iter = iter + 1$.
(7)  Output the calculation results.

The flow chart of the improved ant colony algorithm is shown in Figure 5.

### 3.2. Delivery Path Algorithm for Immediate Customer Demand Phase

In this paper, we study the optimization of a cold-chain logistics distribution path considering carbon emission and immediate customer demand [30]. Because of the peak season of cold-chain goods demand, holiday promotion, and other unexpected situations, there may be a delay in the delivery time requested by customers or new customers' orders during the cold-chain distribution process [31,32]. Generally speaking, the immediate customer demand studied in this paper includes new customer demand and changes in the existing customer time window [33].

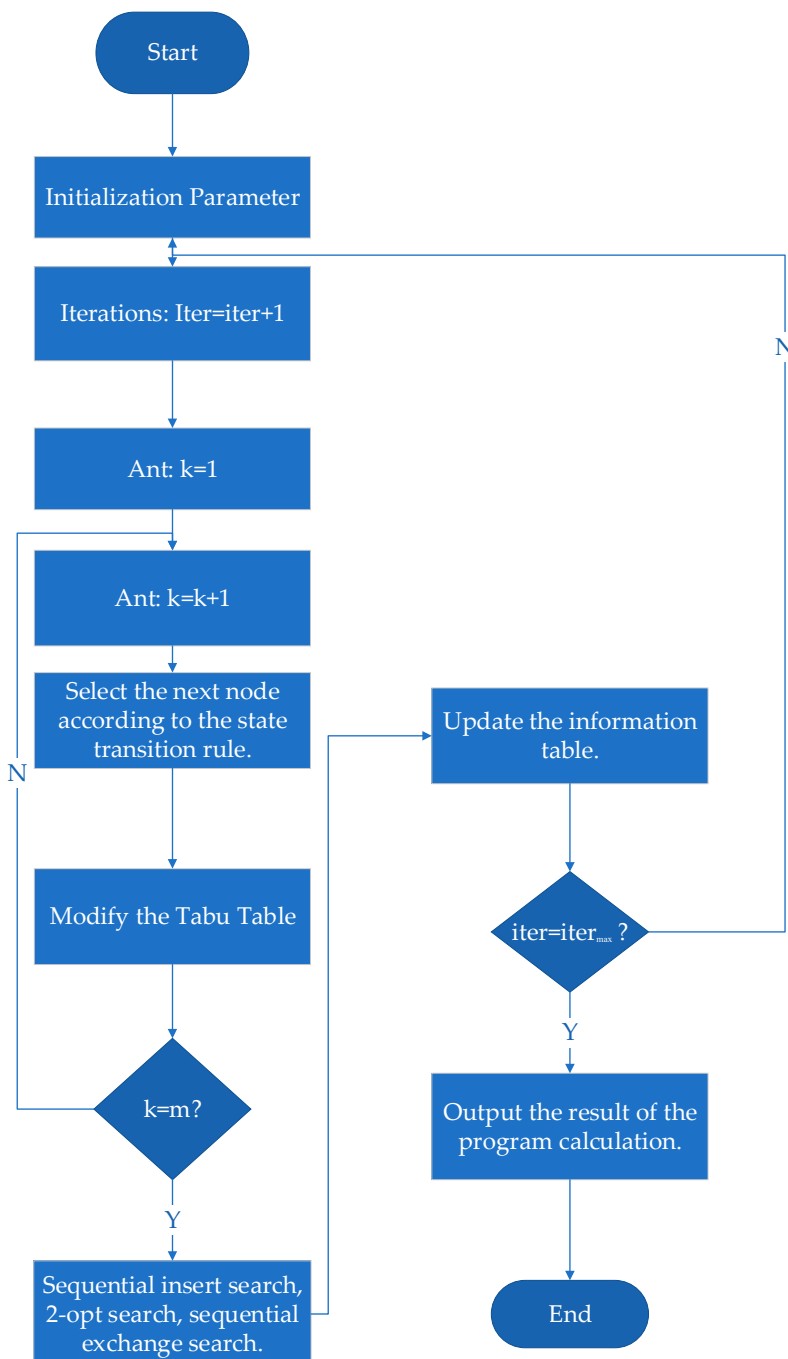

**Figure 5.** Flow chart of improved ant colony algorithm.

When the distribution center receives information on immediate customer demand, it simplifies the complex problem by converting the dynamic information into several static pieces. When the immediate customer demand arises, there are usually two ways to process it immediately and in batches [34,35]. Due to the limitation of immediate processing, the distribution center cannot arrive at the optimal route when the distribution center has a heavy workload, so this paper selects the timing processing method in batch processing to process the immediate customer demand. In the actual distribution, the distribution center needs to divide the products in advance according to the customer's order, so local optimization is more relevant to reality [36,37].

For the new customer demand, the vehicle that has already departed from the distribution center will not be considered to return to the distribution center halfway. The new

customer demand can be delivered by the refrigerated vehicle that has yet to depart [38,39]. Moreover, the distribution center will arrange a new vehicle if there is no eligible vehicle. When the time window for the original customer changes, the distribution center will make local optimization adjustments to the distribution route at the timing decision point and communicate the planned new route to the distribution drivers [40,41].

Compared with other methods, the insertion method can obtain the results in a shorter time so that the immediate customer demand can be satisfied and the timeliness of the adjusted and optimized distribution plan can be ensured. The flow chart of the insertion algorithm is shown in Figure 6.

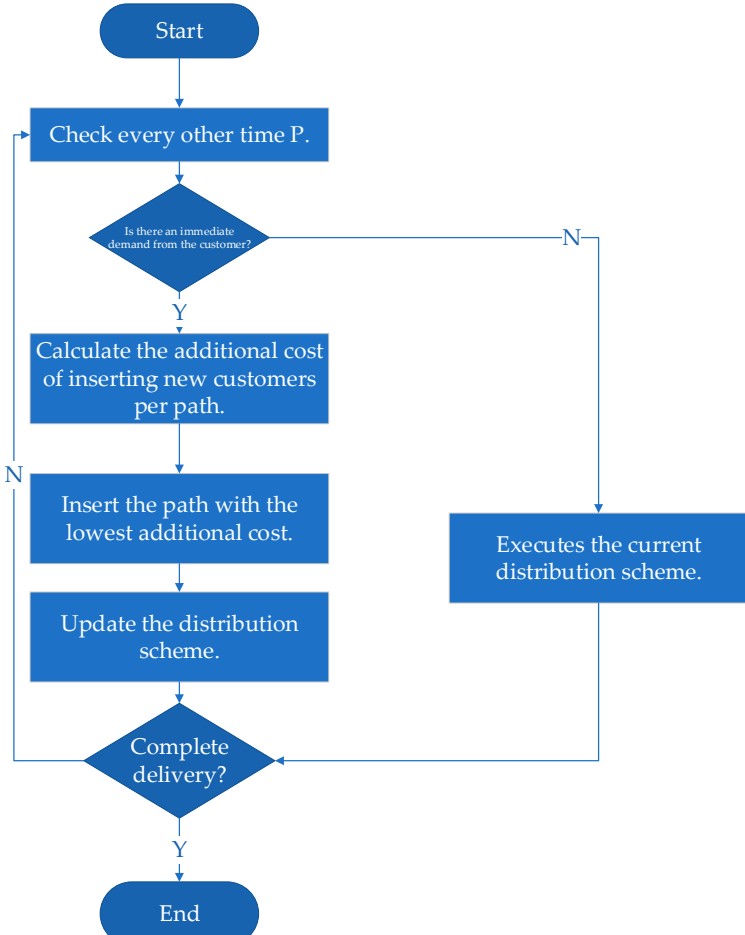

**Figure 6.** Flow chart of insertion algorithm.

In order to solve the optimization model of the cold-chain logistics distribution path considering carbon emission and the customer's immediate demand better, this paper selects an improved ant colony algorithm to solve the initial planning stage. The insertion algorithm solves the optimization phase of the customer's instant demand. The overall algorithm design is shown in Figure 7.

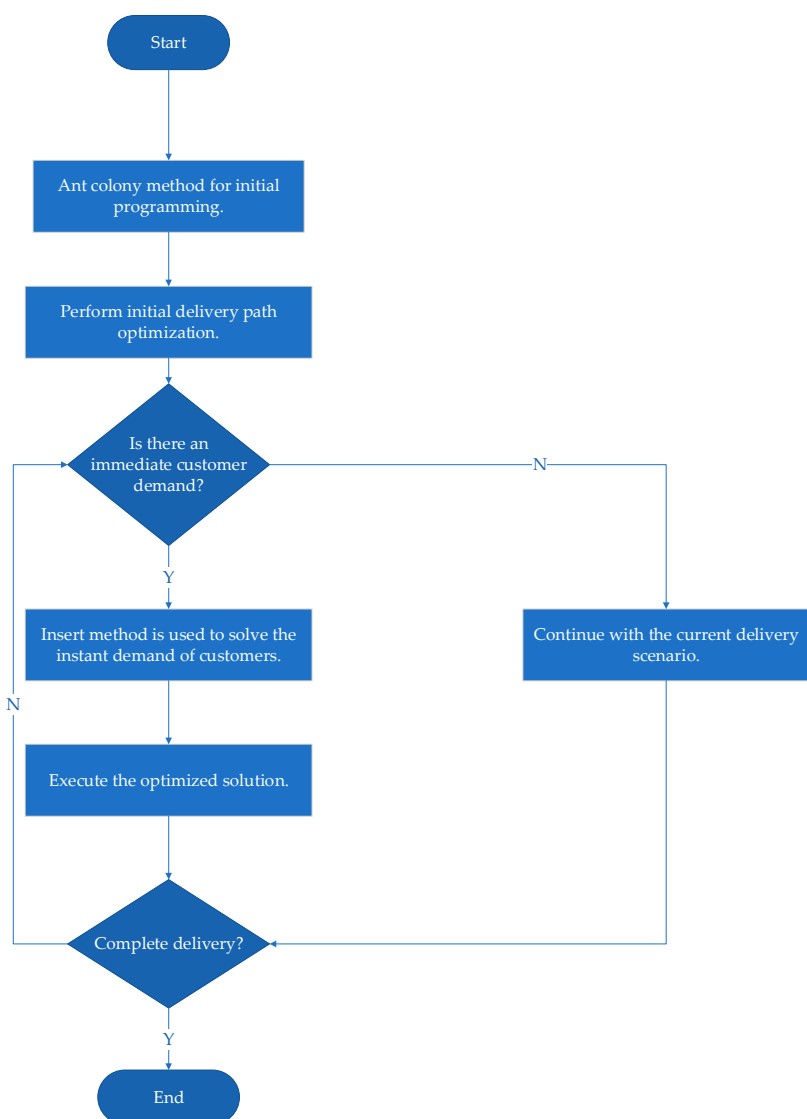

**Figure 7.** Overall algorithm design diagram.

## 4. Example Analysis of the Calculation

*4.1. Background Analysis and Parameter Setting*

Enterprise M is a large fresh food distribution company in Shaanxi province. The services include meat, fresh fruits, vegetables, and aquatic products. Moreover, other work on all customer orders is received the day before at 3:30, and the distribution vehicles depart from the distribution center at 5:00. New customer requirements are received by the distribution center between 3:50 and 4:30. We can add the customer's new requirements to the initial delivery route. Assuming that the departing vehicles can meet the distribution needs of new customers, we will replan the distribution route. If the new demand cannot be met, we will arrange another car for distribution. In addition, the distribution center will also receive the customer's request for service time window adjustment, which requires local optimization of the distribution route. This paper obtains the linear distance between the distribution center and the customer through the scale. The basic information about the initial customers is shown in Table 2.

**Table 2.** Initial customer demand information.

| Customer Number | Longitude (E) | Latitude (N) | Demand (t) | Service Time (h) | ET | et | lt | LT |
|---|---|---|---|---|---|---|---|---|
| 0 | 108.677091 | 34.266719 | - | - | - | - | - | - |
| 1 | 108.834904 | 34.298973 | 0.9 | 0.36 | 123 | 168 | 192 | 224 |
| 2 | 108.351339 | 34.457663 | 1.2 | 0.48 | 111 | 159 | 181 | 193 |
| 3 | 108.659804 | 34.335292 | 0.6 | 0.24 | 24 | 57 | 92 | 114 |
| 4 | 108.401643 | 34.201826 | 1 | 0.4 | 15 | 36 | 54 | 121 |
| 5 | 108.431812 | 34.426692 | 0.5 | 0.2 | 113 | 142 | 154 | 171 |
| 6 | 108.686791 | 34.458481 | 1 | 0.4 | 96 | 133 | 170 | 208 |
| 7 | 108.534434 | 34.336947 | 0.5 | 0.2 | 50 | 78 | 96 | 113 |
| 8 | 108.733897 | 34.315858 | 0.7 | 0.28 | 85 | 115 | 148 | 193 |
| 9 | 108.810638 | 34.422681 | 0.6 | 0.24 | 12 | 45 | 89 | 121 |
| 10 | 108.163187 | 34.358199 | 1.3 | 0.52 | 102 | 144 | 195 | 238 |
| 11 | 108.613462 | 34.124683 | 1.1 | 0.44 | 23 | 69 | 194 | 114 |
| 12 | 108.917593 | 34.157497 | 0.6 | 0.24 | 15 | 42 | 68 | 92 |
| 13 | 108.411762 | 34.306202 | 0.7 | 0.28 | 18 | 33 | 50 | 68 |
| 14 | 108.342154 | 34.367553 | 0.7 | 0.28 | 103 | 136 | 158 | 169 |
| 15 | 108.194996 | 34.259948 | 0.9 | 0.36 | 46 | 87 | 137 | 162 |
| 16 | 108.939718 | 34.202915 | 0.8 | 0.32 | 24 | 53 | 88 | 105 |
| 17 | 108.833701 | 34.251392 | 0.6 | 0.24 | 124 | 149 | 183 | 207 |
| 18 | 108.977323 | 34.398478 | 0.8 | 0.32 | 117 | 132 | 156 | 172 |
| 19 | 108.601433 | 34.395234 | 0.7 | 0.28 | 52 | 78 | 95 | 112 |
| 20 | 108.957505 | 34.346586 | 0.5 | 0.2 | 21 | 45 | 92 | 135 |
| 21 | 108.453489 | 34.154947 | 0.6 | 0.24 | 18 | 26 | 35 | 42 |
| 22 | 108.904797 | 34.360362 | 0.7 | 0.28 | 124 | 159 | 180 | 208 |
| 23 | 108.760642 | 34.507643 | 0.5 | 0.2 | 16 | 32 | 68 | 135 |
| 24 | 108.615176 | 34.532539 | 0.7 | 0.28 | 45 | 87 | 106 | 182 |
| 25 | 109.021785 | 34.273477 | 0.7 | 0.28 | 42 | 93 | 149 | 181 |
| 26 | 108.339864 | 34.218985 | 1.3 | 0.52 | 18 | 25 | 47 | 60 |
| 27 | 108.234023 | 34.146337 | 0.8 | 0.32 | 21 | 36 | 59 | 113 |
| 28 | 108.961859 | 34.267609 | 0.6 | 0.24 | 58 | 97 | 142 | 173 |
| 29 | 108.023154 | 34.131234 | 0.7 | 0.28 | 54 | 72 | 103 | 124 |
| 30 | 108.580261 | 34.458556 | 0.5 | 0.2 | 58 | 94 | 139 | 192 |
| 31 | 108.726415 | 34.356136 | 1.1 | 0.44 | 21 | 32 | 48 | 64 |
| 32 | 108.281024 | 34.328557 | 0.6 | 0.24 | 23 | 42 | 61 | 83 |
| 33 | 108.079304 | 34.282961 | 1.1 | 0.44 | 26 | 47 | 68 | 107 |
| 34 | 108.904597 | 34.446687 | 0.7 | 0.28 | 48 | 98 | 148 | 174 |
| 35 | 109.010218 | 34.444346 | 1.2 | 0.48 | 26 | 63 | 91 | 142 |
| 36 | 108.077109 | 34.246373 | 0.5 | 0.2 | 169 | 184 | 207 | 256 |
| 37 | 108.136842 | 34.526286 | 0.7 | 0.28 | 108 | 121 | 147 | 163 |
| 38 | 108.885776 | 34.523326 | 0.6 | 0.24 | 21 | 68 | 91 | 115 |
| 39 | 109.010997 | 34.185556 | 0.7 | 0.28 | 18 | 25 | 59 | 96 |
| 40 | 109.056321 | 34.246891 | 1 | 0.4 | 76 | 98 | 135 | 164 |
| 41 | 109.137439 | 34.356827 | 0.6 | 0.24 | 23 | 48 | 89 | 154 |
| 42 | 108.718379 | 34.126482 | 0.8 | 0.32 | 113 | 143 | 165 | 201 |
| 43 | 108.107532 | 34.415861 | 0.9 | 0.36 | 91 | 105 | 134 | 148 |

The immediate customer demand involved in enterprise M is divided into two categories, one for new customer needs, denoted by 0, and the other for adjustments to the original customer time window, denoted by 1. The information on the immediate customer demand of enterprise M is shown in Table 3.

**Table 3.** Customers instantly demand information.

| Immediate Demand Type | Customer Number | Receiving Moment | Longitude (E) | Latitude (N) | Demand (t) | Service Time (h) | ET | et | lt | LT |
|---|---|---|---|---|---|---|---|---|---|---|
| 0 | 44 | 3:54 | 108.613742 | 34.426692 | 0.8 | 0.32 | 84 | 125 | 153 | 201 |
| 0 | 45 | 4:12 | 108.709566 | 34.245249 | 0.6 | 0.24 | 149 | 172 | 196 | 256 |
| 1 | 22 | 5:13 | 108.904797 | 34.360362 | 0.7 | 0.28 | 35 | 35 | 50 | 50 |
| 1 | 18 | 5:18 | 108.977323 | 34.398478 | 0.8 | 0.32 | 137 | 137 | 148 | 148 |

The relevant parameters are designed as follows: the distribution center is the same type of refrigerated vehicle, the maximum vehicle load $Q$ is 5t, the fixed cost of the refrigerated vehicle $f_k$ is 350 yuan, and the transportation cost of the refrigerated vehicle $F_k$ is 8 yuan/km; M enterprise is set according to the actual situation of the enterprise $C_e = 100$ yuan, $C_l = 150$ yuan; the fuel consumption of the vehicle empty $\rho^0$ and $\rho^*$ full load unit distance and is 0.35 L/km and 0.7 L/km, respectively; the energy consumption of refrigeration equipment is 0.00868/kg km, and the unit carbon tax $p_c$ is 100 yuan/t; the $CO_2$ emission factor $e$ is 2.61 kg/L. This paper assumed that the outdoor temperature is constant at 18 °C. The temperature inside the refrigerated vehicle compartment is set at 0 °C, and the cooling cost of the logistics vehicle during the transportation is 35 yuan/h. The temperature inside the compartment increases by 3 °C when the distribution point carries out unloading, the additional cooling cost due to the temperature difference is 6 yuan (°C/h), and the speed of the distribution vehicle is 60 km/h. The basic parameters of the improved ant colony algorithm are: $\alpha = 1$, $\beta = 2$, $\rho = 0.75$, $Q = 100$, and $\theta = 2$, the number of ants $m = 40$, and the maximum number of iterations $iter_{max} = 200$ [22]. In addition, the weight of the lowest transportation cost is 0.75, and the lowest customer dissatisfaction is 0.25.

*4.2. Analysis of Results*

(1)　Solving under the initial distribution demand

The distribution information and related parameters of customers are imported. The initial distribution model is constructed and improved by the design of the ant colony algorithm, which is solved by MATLAB software. The iteration curve of the total cost can be obtained, as shown in Figure 8. After 60 iterations, the total distribution cost curve is more stable, and the optimal value of the total distribution cost is obtained when the number of iterations is 200.

The algorithm is run 200 times to obtain the optimal results. According to the results, it is known that the cold-chain distribution center needs to send eight refrigerated trucks to serve 43 customers, and the optimal roadmap for refrigerated truck distribution is shown in Figure 9.

The vehicle distribution routes under the improved ant colony algorithm can be seen in Figure 9, the vehicle numbers are represented by vehicles 1–8, and each vehicle's specific distribution tasks and customers are shown in Table 4.

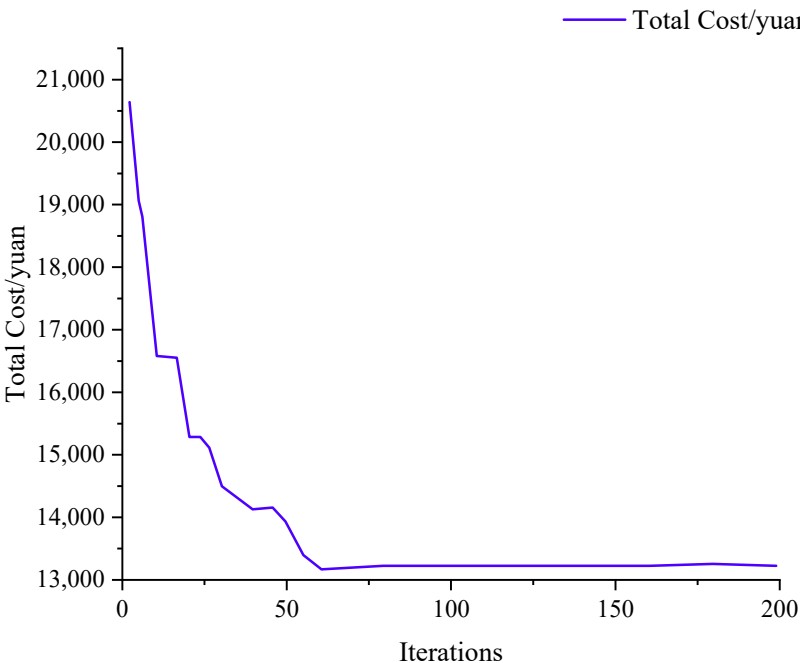

**Figure 8.** Algorithm convergence diagram of total distribution cost.

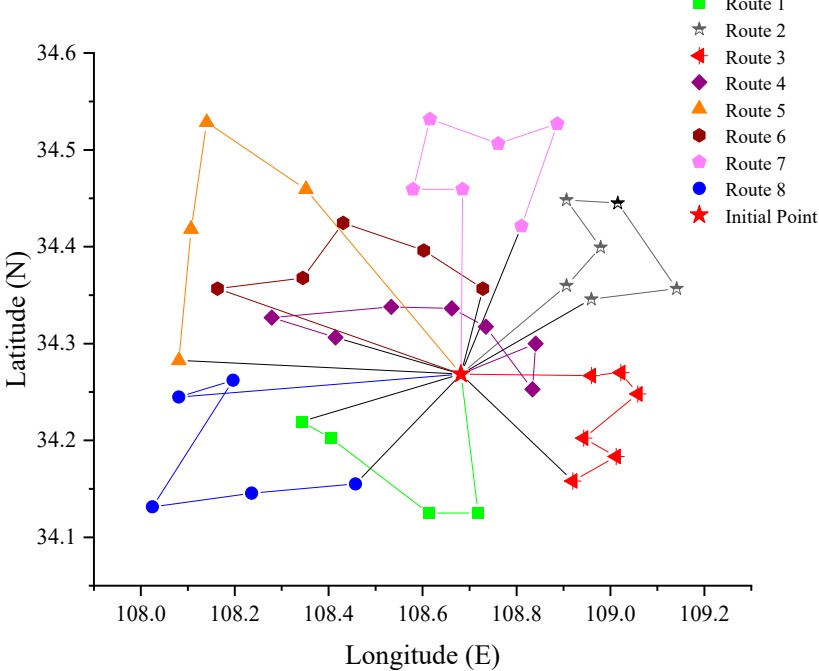

**Figure 9.** Optimal route map.

As shown in Table 4, enterprise M needs eight vehicles to complete all distribution tasks. The first vehicle distribution route is 0-26-4-11-42-0, with a total load rate of 84%. The second vehicle distribution route is 0-20-41-35-34-18-22-0, with a total load rate of 90%. The third vehicle distribution route is 0-12-39-16-40-25-28-0, with a total load rate of 88%. The fourth vehicle delivery route is 0-13-32-7-3-8-17-1-0, with a total load rate of 92%. The fifth vehicle delivery route is 0-33-43-37-2-0, with a total load rate of 78%. The sixth vehicle delivery route is 0-31-19-5-14-10-0, with a total load rate of 86%. The seventh vehicle delivery route is 0-9-38-23-24-30-6-0, with a total load rate of 78%. The eighth vehicle delivery route is 0-21-27-29-15-36-0, with a total load rate of 70%.

**Table 4.** Specific distribution routes of vehicles.

| Vehicle | Route | Vehicle Load (t) | Vehicle Full Load Ratio |
|---|---|---|---|
| 1 | 0-26-4-11-42-0 | 4.2 | 84% |
| 2 | 0-20-41-35-34-18-22-0 | 4.5 | 90% |
| 3 | 0-12-39-16-40-25-28-0 | 4.4 | 88% |
| 4 | 0-13-32-7-3-8-17-1-0 | 4.6 | 92% |
| 5 | 0-33-43-37-2-0 | 3.9 | 78% |
| 6 | 0-31-19-5-14-10-0 | 4.3 | 86% |
| 7 | 0-9-38-23-24-30-6-0 | 3.9 | 78% |
| 8 | 0-21-27-29-15-36-0 | 3.5 | 70% |

This paper improves customer satisfaction as the premise for customer delivery. Customer satisfaction is expressed by a trapezoidal fuzzy affiliation function time window to minimize delivery costs and maximize customer satisfaction. After calculation, the results of each distribution vehicle solution under the initial demand are shown in Table 5.

**Table 5.** Initial demand distribution solution results.

| Vehicles | Vehicle Fixed Costs | Vehicle Transportation Costs | Temperature Costs | Carbon Costs | Time Window Penalty Costs | Total Customer | Dissatisfaction |
|---|---|---|---|---|---|---|---|
| 1 | 350 | 685.36 | 139.98 | 184.66 | 58.5 | 1417.49 | 24.54% |
| 2 | 350 | 871.60 | 158.95 | 259.58 | 0 | 1640.14 | 12.26% |
| 3 | 350 | 689.12 | 143.53 | 187.99 | 0 | 1370.64 | 16.86% |
| 4 | 350 | 887.68 | 162.25 | 284.09 | 166.5 | 1850.52 | 22.10% |
| 5 | 350 | 1123.57 | 164.61 | 283.91 | 0 | 1922.09 | 15.29% |
| 6 | 350 | 916.35 | 157.98 | 160.03 | 0 | 1584.36 | 19.44% |
| 7 | 350 | 797.08 | 140.80 | 195.84 | 0 | 1483.72 | 13.69% |
| 8 | 350 | 1210.74 | 162.48 | 208.09 | 91 | 2022.32 | 24.87% |

Through path optimization, the total driving distance of distribution was 897.69 km, and the total distribution cost was 13,291.27 yuan, including vehicle fixed cost of 2800 yuan, accounting for 21.07%. Vehicle transportation cost was 17,181.50 yuan, accounting for 54.03%. The temperature cost was 1229.58 yuan, accounting for 9.25%. Carbon emission cost was 1764.19 yuan, accounting for 13.27%. The time window penalty cost was 316 yuan, accounting for 2.38%. Among the costs, the most significant proportion is vehicle transportation, which M enterprise should focus on. The minor proportion is the time window penalty cost, and the average customer dissatisfaction is 18.63%, proving that the delivery time of distribution vehicles conforms to the soft time window limit and the high customer satisfaction.

(2)　Problem solving under immediate customer demand

The distribution center of enterprise M updates the customer demand information every 20 min during the period of 3:50–4:30, and the new demand received from customers can be added to the initial distribution route. After distribution started at 5:00, the outstanding distribution routes were updated every 20 min in response to customer time window adjustments. Customer 44 made a delivery request to the M enterprise distribution center at 3:54, and customer 45 made a delivery request at 4:12. Then, customer 22 made an urgent purchase request to the distribution center of enterprise M at 5:13. Customer 18 requested an early delivery of goods to the distribution center of enterprise M at 5:18. The distribution path under immediate customer demand is updated using the insertion method. The results are shown in Table 6.

**Table 6.** Update results of distribution routes under immediate customer demand.

| Immediate Demand Type | Customer Number | Receiving Moment | Vehicle | Distribution Cost (yuan) | Adjusted Distribution Path |
|---|---|---|---|---|---|
| 0 | 44 | 3:54 | 7 | 1576.46 | 0-9-38-23-24-30-44-6-0 |
| 0 | 45 | 4:12 | 1 | 1472.39 | 0-26-4-11-42-45-0 |
| 1 | 22 | 5:13 | 2 | 1897.15 | 0-20-22-41-35-18-34-0 |
| 1 | 18 | 5:18 | 2 | 1897.15 | 0-20-22-41-35-18-34-0 |

At 4:10, the distribution center updated the distribution route. Customer 44 had a new distribution demand. The distribution route of vehicle 7 was changed, the load capacity of vehicle 7 was changed from 3.9 to 4.7 t, the total distribution cost was 1576.46 yuan, and the additional cost was 92.74 yuan.

At 4:30, the distribution center updated the distribution route. Customer 45 had a new distribution demand. The distribution route of vehicle 1 was changed, the load capacity of vehicle 1 changed from 4.2 to 4.8 t, the total distribution cost was 1472.39 yuan, and the additional cost was 54.90 yuan.

The distribution center received customers' requests 22 and 18 to adjust the service time window at 5:13 and 5:18, respectively. The distribution center updated and re-optimized the distribution route at 5:20 when refrigerated vehicle 2 was on its way to customer 20. The distribution center made adjustments to optimize the customers' points on path two that were not delivered after customer 20. The distribution cost of route two after adjustment was 1897.15 yuan, which increased the additional cost by 257.01 yuan.

As shown in Table 7, the loading rate of vehicle 1 changes from 84% to 96%, and that of vehicle 7 increases from 78% to 94%, saving M enterprises' resources while increasing the loading rate and satisfying the needs of additional customers. Vehicle 2 adds a cost of 257.01 yuan but satisfies the immediate demand of customers 18 and 22, avoiding more penalty costs.

**Table 7.** Distribution vehicle loading rate.

| | Vehicle 1 | Vehicle 2 | Vehicle 3 | Vehicle 4 | Vehicle 5 | Vehicle 6 | Vehicle 7 | Vehicle 8 |
|---|---|---|---|---|---|---|---|---|
| Total load rate of distribution vehicles under initial demand | 84% | 90% | 88% | 92% | 78% | 86% | 78% | 70% |
| Total load rate of delivery vehicles under immediate customer demand | 96% | 90% | 88% | 92% | 78% | 86% | 94% | 70% |

## 5. Conclusions

Based on the cold-chain logistics, this paper adopts the improved ant colony algorithm to solve the initial distribution path and the insert method to solve the immediate demand of customers, aiming at the minimum distribution cost and the maximum customer satisfaction [42,43]. Using M enterprise's actual data as an example, this paper uses MATLAB software to obtain the optimal distribution path. It constructs a trapezoidal fuzzy membership function to express the relationship between customer satisfaction and the time window and local optimization of the distribution path [44]. In addition, this paper also considers the carbon emissions in the total cost to local optimization of the distribution path. Based on the conclusion of this paper, we put forward the following suggestions for developing low-carbon cold-chain logistics.

Firstly, optimize the distribution plan of M enterprise. Choosing the optimal distribution path is a complex problem for M enterprise. According to M enterprise's route optimization result, the distribution distance dramatically affects the distribution cost. According to M enterprise's route optimization result, the distribution distance dramatically

affects the distribution cost, which requires rationalizing routes and reducing vehicle travel time. Second, carbon emissions are related to vehicle loads, which can be delivered to high-demand customers first, thereby reducing the cost of carbon emissions. In addition, M enterprise should also pay attention to the daily maintenance of refrigerated vehicles to ensure that refrigerated vehicles have better vehicle driving and cooling capacity to reduce fuel consumption and carbon emissions while maintaining refrigeration and reducing vehicle costs. The most important thing is to measure the cost and customer satisfaction of distribution to ensure the long-term development of M enterprise.

Second, strengthen the service consciousness of M enterprise. Customer satisfaction is crucial to the business and includes both satisfaction with product quality and delivery services. For the distribution center, it includes ensuring the freshness of products and providing customer satisfaction with the service. For M enterprise, we should take "3Q" as the primary service idea, standardize the delivery procedure, reduce the operation time of the intermediate link as much as possible, and respond to the customer's demand in time. Start with every link of the cold-chain logistics to improve the quality of the delivery service and ensure the freshness and customer satisfaction of the products.

Third, improve the level of logistics distribution information. M enterprise must infiltrate intelligence into each link of distribution. In the distribution of products for the whole process, monitoring speeds up the operation process and improves distribution accuracy and security. It can also respond to customers' real-time needs and improve m enterprise's ability to deal with customers' emergencies, such as drivers' sudden health problems and refrigerated vehicles' sudden malfunctions, which cannot be predicted, so how M enterprise deals with these emergencies is very important. It is suggested that M enterprise install a management system and intelligent positioning in the refrigerated vehicle, know and forecast traffic status in time, respond to customer demand quickly, and improve its core competitiveness.

This research is helping to improve customer satisfaction and reduce carbon emissions in cold-chain logistics distribution [45]. It is of great practical significance to realize the maximization of environmental and economic benefits. In addition, this study adapts to the development of the current era and meets the requirements of low-carbon sustainable development of cities [46]. Solving the uneconomic and unsustainability problem in cold-chain logistics distribution is helpful, and it promotes the reform of enterprises. Optimizing cold-chain logistics distribution helps enhance the clustering and network connectivity of logistics resources and ensures the capacity and efficiency of cold-chain logistics.

Although this paper has some reference value for studying cold-chain distribution path optimization, the optimization problem considers the carbon emission and the customers' immediate demand. Due to the limitations of assumptions and constraints, there are some improvements in this paper's speed and temperature distribution.

**Author Contributions:** Conceptualization, Z.M. and H.W.; methodology, J.Z., Z.M. and H.W.; software, J.Z. and S.G.; validation, Z.M., J.Z. and H.W.; formal analysis, J.Z. and H.W.; investigation, J.Z. and H.W.; resources, Z.M. and H.W.; data curation, J.Z. and H.W.; writing—original draft preparation, J.Z. and H.W.; writing—review and editing, J.Z. and H.W.; visualization, J.Z. and H.W.; supervision, J.Z. and H.W.; project administration, Z.M.; funding acquisition, Z.M. All authors have read and agreed to the published version of the manuscript.

**Funding:** This research was funded by the National Natural Science Foundation of China, grant number 71871084 and Social Science Grand Research of the Hebei Education Department, grant number BJ2021070.

**Institutional Review Board Statement:** Not applicable.

**Informed Consent Statement:** Not applicable.

**Data Availability Statement:** Data are contained within the article.

**Acknowledgments:** All authors gratefully acknowledge the comments of reviewers and editors of this article.

**Conflicts of Interest:** The authors declare no conflict of interest.

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
