# Peer review of "Optimization of Sustainable Bi-Objective Cold-Chain Logistics Route Considering Carbon Emissions and Customers’ Immediate Demands in China"

_sustainability, doi:10.3390/su15075946_

Round 1

Reviewer 1 Report

pag. line 

2 67          "The experimental results show that the optimal parameter combination." not understandable

4   148-149     "Supposition 1: There is only one distribution center and distribution vehicles need to start and finish the routes in it." better understandable

5   Table 1     "The length of time it takes ..." replace by "The time it takes"

5   Table 1     "The cos factor for product ...." replace by "The cost factor for product ...."

5 180 Equation (3) shouldn't it have a negative coefficient, due to the temperature being lowered, to preserve the products?

8   Figure 4    what is the unit of time ?

10 319 should be explained or referenced bibliography to be consulted

10 342 "Zij(t) is chaotic", should be better explained or referred to bibliography where it is explained

10 350 the variables should be identified and better explained or referred to bibliography where they are explained

11 363-364 it would be better "3) Sequential insertion strategy: Insert customer points in the route or in different routes."

11 372 "...equation 26,..." should be "...equation (26),..."  should be explained or referenced bibliography to be consulted

11 372 "... table tabu k ..." 

11 378 "...formula 29..." should be "...formula (29)..." 

12 figure could be better done without crossing lines

figure does not have the same steps as the previous description

14 436-438 rewrite the sentence to be better understandable

15 Table 2 only Service time indicates time units. In the other variables, the units of time should also be indicated

16 468-469 Ant colony algotithm should be explained even briefly or referenced bibliography to be consulted

16 477 MATLAB software, it should be better explained how it was used, commands, libraries, etc.

17 Figure 8 axes in english

17 Figura 9 each route its color would look better

20 574-575 rewrite the sentence to be better understandable

shouldn't the way to refer to the bibliography be between square brackets?

Do numbers in superscript refer to bibliographical references?

It would be good if there was a comparison between the results obtained with this work, and somehow the way it was done previously.

Author Response

According to your valuable comments, we have revised the article. The details are in "Response to Reviewer 1 Comments". Thank you again for your valuable advice.

Reviewer 2 Report

The authors propose an interesting topic and do some research work. However, the research work has the following issues that should be addressed by the authors.

1. Introduction section

1) Please define the perishable nature of cold-chain products (Lines 36 and 37) and explain why such a nature can cause the increase of the carbon emission.

2) The motivations of considering the real-time demand of customers (Lines 39 and 40) seem not to be related to the carbon emissions. So the authors need to indicate the reasons that they formulate such kind of demands in the routing.

3) There are “two-objective” (in the title) and “bi-objective” (Line 42). In the literature, the latter is most commonly used.

4) The literature review is poorly presented:

- In the routing problem, there are three widely used approaches to reduce the carbon emissions, i.e., carbon tax regulation (authors’ adoption), carbon trading regulation (see Liu et al., https://doi.org/10.1016/j.resconrec.2020.104715) and multi-objective optimization (see Demir et al., https://doi.org/10.1080/00207543.2019.1620363). The literature review on green routing should cover them and indicate which approach the reviewed article adopts.

In this case, the authors need to clarify why they use the cost of carbon emissions in the vehicle routing model.

- The routing problem uses time windows to improve the timeliness of the transportation that is related to the customer satisfaction. As indicated by Sun et al. (https://doi.org/10.1080/00207543.2019.1620363), there are hard time windows, soft time windows and fuzzy soft time windows. When reviewing articles, the authors need to present the kind of time windows that the article uses.

Similarly, the authors need to explain the considerations why they use fuzzy soft time windows, i.e., give the advantages of such time windows, which can refer to Sun et al.’s work.

- After reviewing the existing relevant studies, the authors need to present the weaknesses of these studies and explain how their work can fix these weaknesses in detail. Then, the authors need to summarize their contributions.

5) The authors need to explain why heuristic algorithm is suitable to solve the problem. NP hardness of the vehicle routing problem is never mentioned by the authors. What are “heuristic function and pheromone”? Why should the ant colony algorithm be improved by the authors’ strategies?

2. Problem description and modeling

1) Please define the symbols before showing the mathematical model. It is better to classify them into sets, indices, parameters and variables.

2) Are there any references to support Assumption 2, since it is rare to see “the number of distribution vehicles is sufficient” in the literature.

3) I’m wondering why both soft time windows and fuzzy soft time windows exist in the model.

4) My major concern is that λi is also a variable and should be also defined in Table 1. In the mathematical model, there should be equations calculating λi. Therefore, the authors need to carefully if check their model is correct.

5) The type of the model should be indicated (especially, linear or nonlinear), since it is related to the selection of algorithms.

3. Algorithm design

The main problem of the algorithm design is that the description is too general. Please connect the algorithm design with the specific vehicle routing problem.

4. Example analysis of the calculation

1) Could the authors provide some comparisons to indicate the effectiveness of their algorithm?

2) Figure 8 contains Chinese characteristics, which should be avoided.

3) Since a bi-objective optimization is given by the authors, the Pareto frontier should be included in the experiment results.

4) Sensitivity analysis of the optimization results with respect to the carbon emission cost rate is recommended, since there is evidences that the routing results are not sensitive to the emission cost rate (see Sun et al., https://doi.org/10.1155/2018/8645793).

5) Please summarize the findings from the experiment and propose some managerial insights that can help decision makers to efficiently organize the cold chain deliveries.

Author Response

According to your valuable comments, we have revised the article. The details are in "Response to Reviewer 2 Comments". Thank you again for your valuable advice.

Round 2

Reviewer 2 Report

The revisions provided by the authors are fine.